# Unique transcriptome signatures and GM-CSF expression in lymphocytes from patients with spondyloarthritis

M.H. Al-Mossawi [1], L. Chen[1], H. Fang[2], A. Ridley[1], J. de Wit[1], N. Yager[1], A. Hammitzsch[1], I. Pulyakhina[2], B.P. Fairfax[2], D. Simone[1], Yao Yi[1], S. Bandyopadhyay[1], K. Doig[1], R Gundle[1], B. Kendrick[1], F. Powrie[3], J.C. Knight[2] & P. Bowness[1]

Spondyloarthritis encompasses a group of common inflammatory diseases thought to be driven by IL-17A-secreting type-17 lymphocytes. Here we show increased numbers of GM-CSF-producing CD4 and CD8 lymphocytes in the blood and joints of patients with spondyloarthritis, and increased numbers of IL-17A+GM-CSF+ double-producing CD4, CD8, γδ and NK cells. GM-CSF production in CD4 T cells occurs both independently and in combination with classical Th1 and Th17 cytokines. Type 3 innate lymphoid cells producing predominantly GM-CSF are expanded in synovial tissues from patients with spondyloarthritis. GM-CSF+CD4+ cells, isolated using a triple cytokine capture approach, have a specific transcriptional signature. Both GM-CSF+ and IL-17A+GM-CSF+ double-producing CD4 T cells express increased levels of GPR65, a proton-sensing receptor associated with spondyloarthritis in genome-wide association studies and pathogenicity in murine inflammatory disease models. Silencing GPR65 in primary CD4 T cells reduces GM-CSF production. GM-CSF and GPR65 may thus serve as targets for therapeutic intervention of spondyloarthritis.

[1] Botnar Research Centre, Nuffield Department of Orthopaedics, Rheumatology and Musculoskeletal Sciences, University of Oxford, Oxford OX3 7LD, UK. [2] Wellcome Trust Centre for Human Genetics, Nuffield department of medicine, University of Oxford, Old Road campus, Oxford OX3 7BN, UK. [3] Kennedy Institute of Rheumatology, Nuffield Department of Orthopaedics, Rheumatology and Musculoskeletal Sciences, University of Oxford, Oxford OX3 7FY, UK. Correspondence and requests for materials should be addressed to M.H.A.-M. (email: Hussein.al-mossawi@ndorms.ox.ac.uk)

Spondyloarthritis encompasses a group of inflammatory diseases with common pathologic and genetic features. These disorders include ankylosing spondyltis (AS), reactive arthritis (ReA), psoriatic arthritis (PsA) and enteropathic arthritis (EA)[1]. The prevalence of spondyloarthritis is at least 0.5% in European and US populations[2,3]. Only one third of patients achieve disease remission with TNF inhibitor therapies, therefore a significant unmet clinical need remains[4–7].

Genetic, functional and therapeutic evidence implicates IL-17-producing cells in the pathogenesis of spondyloarthritis[8–10]. Within the CD4 T cell compartment, Th17 cells can be pleotropic and, in addition to the classical "type 17" cytokines IL-17A, IL-17F and IL-22[11,12], can also produce IFN-γ[13] and GM-CSF[14]. Murine models of inflammation suggest that co-expression of GM-CSF and IL-17A by CD4 T cells marks out a pathogenic subset of Th17 cells[14,15], and neutralisation of GM-CSF in the Sagakuchi model of spondyloarthritis causes improvement in joint inflammation[15]. Noster et al.[16] have provided evidence that GM-CSF and IL-17A are antagonistically regulated, but it is unclear if the CD4/GM-CSF axis is acting within the context of Th17 immunity or independently. Additional pathogenic markers of murine Th17 cells include GPR65[17], a receptor implicated in the pathogenesis of spondyloarthritis by genome-wide association studies[18].

In addition to classical CD4 Th17s, CD161-expressing CD8 T cells[19], γδ T-cells[20], innate lymphoid cells[21] (ILC) and B cells[22] have also been shown to be producers of IL-17A and/or GM-CSF. ILC are broadly divided into three groups; ILC1s express T-bet, ILC2 express GATA3 and ILC3 express RORγt and are thought to be involved in type 17 immunity[23]. ILCs are present in the gut[24], lung[25] and skin[26], and have been described in the joint in psoriatic arthritis[27]. The contribution of ILCs to pathogenic immune responses is still being actively investigated.

Here we show that multiple lymphoid populations (including ILCs) express more GM-CSF in patients with spondyloarthritis. Increased lymphocyte GM-CSF production occurs both in the context of a polyfunctional type 17 response, but also independently of IL-17A. Primary human GM-CSF-producing CD4 cells, isolated using a multiple cytokine capture approach, show a specific transcriptional profile that includes upregulation of GPR65, and we identify a function for this receptor in CD4 lymphocyte GM-CSF production. Therapeutic targeting of GPR65 may potentially attenuate pathogenic T cells while preserving physiological type 17 immunity.

## Results

**GM-CSF+ lymphocytes are increased in spondyloarthritis.** We first performed intracellular cytokine staining of ex vivo peripheral blood mononuclear cells (PBMCs) from 38 patients with axial spondyloarthritis fulfilling the assessment of spondyloarthritis (ASAS) criteria[28], 14 patients with rheumatoid arthritis (RA) matched for CRP, and 17 sex and age matched healthy donors (Supplementary Table 1 shows patient characteristics, Supplementary Fig. 1A shows flow cytometry gating strategy). IL-17A+ CD4 cells (Th17 cells) were increased in spondyloarthritis compared to both control groups (Fig. 1a, b). We then looked within the Th17 compartment for the percentage of cells also expressing GM-CSF (Fig. 1c). We observed a significant increase in these GM-CSF+Th17 cells in spondyloarthritis compared to healthy donors ($p = 0.0031$, Kruskal–Wallis) and RA inflammatory disease controls ($p = 0.0066$, Kruskal–Wallis). IL-17A+GM-CSF+ cells were also increased in CD8, γδ and CD56+ lymphoid subsets in spondyloarthritis (Fig. 1d).

Next we looked at the percentage of cells producing GM-CSF independently of IL-17A. We observed a significant expansion of GM-CSF+IL-17A− cells in both CD4 and CD8 cells in spondyloarthritis compared to the two control populations (Fig. 1e, f), but saw no difference in the CD56 and γδ compartments (Supplementary Fig. 1D, E), and no difference in the percentage of IFN-γ+ cells was seen in the CD4, γδ and CD56 compartments (Supplementary Fig. 1B–E). GM-CSF+ CD4 frequency was not affected by co-existent psoriasis or IBD (Supplementary Fig. 2A). We did however note increased GM-CSF+ CD4 T cells in patients on anti-TNF therapy, but even with the exclusion of these patients from the analysis, GM-CSF+ CD4 cells were still elevated in spondyloarthritis compared to controls (Supplementary Fig. 2B).

Phenotypic analysis showed that single-positive GM-CSF+ CD4+ cells expressed CCR6 and CD161 at similar frequencies to single-positive IFN-γ+CD4+ cells, whilst double-positive IL-17A+GM-CSF+ CD4+ cells more commonly expressed CCR6 and CD161 (similar to single-positive IL-17A+CD4+ cells) (Fig. 1g). We used Boolean gating to delineate the functional overlap of IL-17A, IFN-γ and GM-CSF production by CD4 T cells in healthy controls, spondyloarthritis and RA (Fig. 1h). In all three cohorts, GM-CSF+ CD4 T cells overlapped with both Th1 and Th17 cells but also existed independently of these two effector subsets. In spondyloarthritis the overall GM-CSF pool was expanded both independently and in combination with of Th1 and Th17.

**Spondyloarthritis synovial T cells are enriched for GM-CSF.** Since spondyloarthritis is characterised by joint inflammation, we next studied synovial fluid mononuclear cells (SFMC) from the inflamed joints of 5 patients with peripheral spondyloarthritis (Supplementary Table 2 shows patient characteristics). We compared the cellular sources of GM-CSF in spondyloarthritis PBMCs and SFMCs using time-of-flight mass cytometry (CyTOF) using a panel of 20 surface and 11 intracellular markers. We gated on all live GM-CSF+ events and performed an unbiased clustering analysis of all parameters using viSNE[29] (Supplementary Fig. 3A, C shows the CyTOF gating and validation of CyTOF using flow cytometry). We observed CD4 cells to be the largest population of GM-CSF-producing cell in both spondyloarthritis peripheral blood and synovial fluid, with significant contributions from CD8 and CD56 cells (Supplementary Fig. 2A, B). Both SFMCs and (matched) PBMCs showed co-production of GM-CSF with IL-17A, IFN-γ, IL-4, IL-10 and TNF (Fig. 2a, b lower sets of panels) confirming and extending our observations in Fig. 1h. We confirmed these data by flow cytometry for 5 additional matched PBMC/SFMC spondyloarthritis samples (Fig. 2c–g). Figure 2c shows not only a marked increase in the size of the CD4 GM-CSF production (>9-fold) in the joint relative to the blood, but also that in this patient 89% of IL-17A+ synovial CD4 T cells co-expressed GM-CSF. In this additional cohort of 5 patients, no significant expansion of IL-17A+ single-positive CD4 cells in the synovial fluid relative to the blood was seen. However, both IL-17A−GM-CSF+ CD4 cells and double-positive IL-17A+GM-CSF+ CD4 and CD8 cells were markedly increased in spondyloarthritis synovial fluid relative to blood (Fig. 2f, g). Additionally, synovial tissue cultures of CD4 T cells from 4 spondyloarthritis patients showed comparable high levels (>30%) of GM-CSF production (Supplementary Fig. 4).

**ILC3 are enriched in the inflamed joint and produce GM-CSF.** The presence of innate lymphoid cells (ILC) in human joints has recently been described in psoriatic arthritis, a form of spondyloarthritis[27], and we next asked if GM-CSF is produced by these cells. We obtained synovial tissue from 4 spondyloarthritis and 3 RA patients undergoing arthroplasty (Supplementary Table 2 shows patient characteristics) and derived synovial tissue

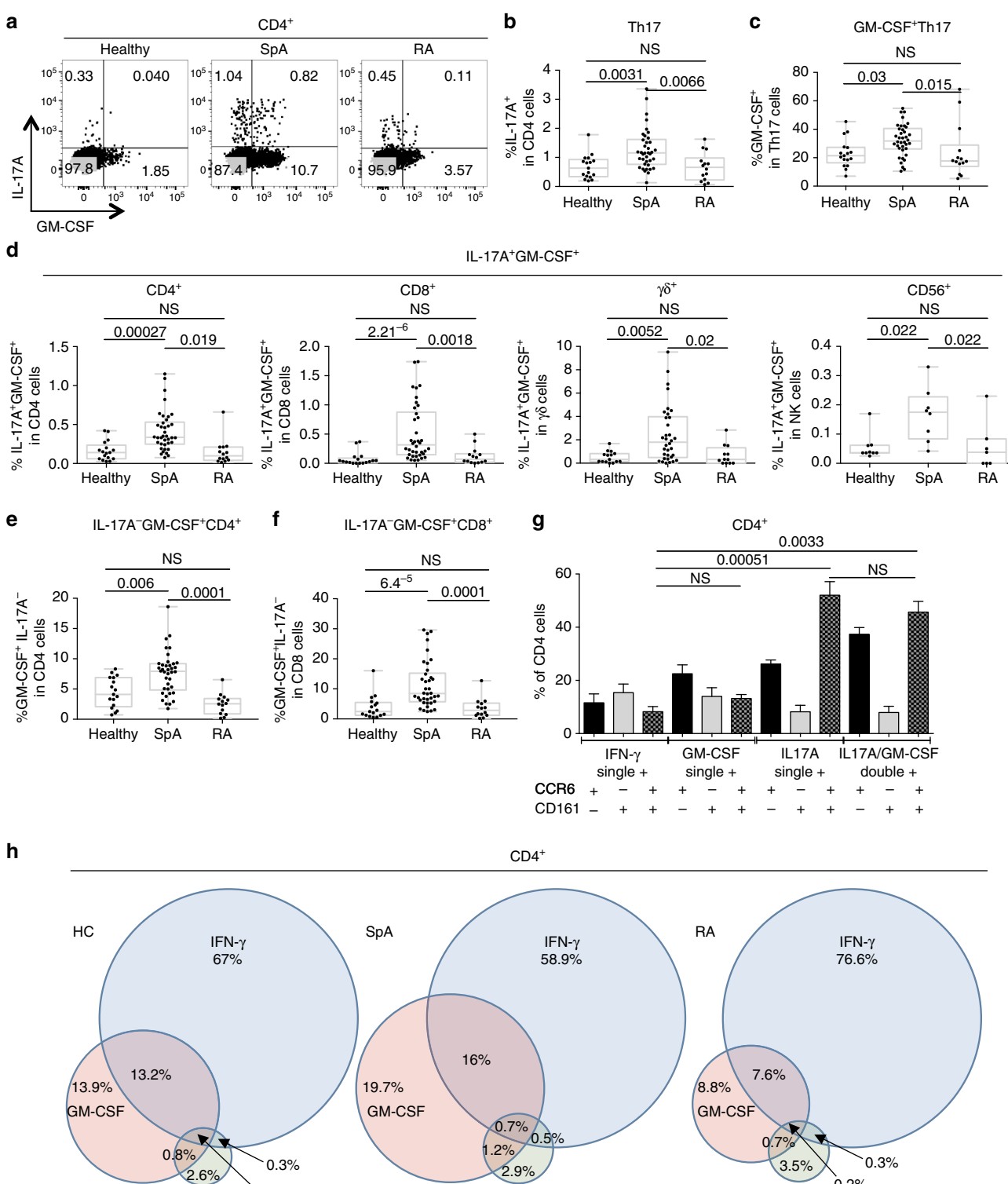

**Fig. 1** GM-CSF+ cells are expanded in multiple lymphoid populations in spondyloarthritis. **a** Representative flow cytometry plots comparing intracellular staining for IL-17A and GM-CSF in healthy, spondyloarthritis (SpA) and rheumatoid arthritis (RA) PBMCs (gated on live CD3+CD4+ lymphocytes). **b** Total IL-17A+CD4+ cells (Th17) and **c** percentage of Th17 cells also expressing GM-CSF for healthy ($n=17$) SpA ($n=38$) and RA ($n=14$). **d** Percentage of IL-17A+GM-CSF+ cells within the CD4+, CD8+, γδ+ and CD56+ gates in the same cohort of patients and controls. Percentage of GM-CSF+ CD4 **e** and CD8 **f** independently of IL-17A in the cohort of patients and controls. **g** CD161 and CCR6 expression was measured by flow cytometry on single positive IL-17A+, IFN-γ+, GM-CSF+ and IL-17A+GM-CSF+ double-positive CD4 cells ($n=8$; mean + SEM shown, Friedman's test). **h** Venn diagram of mean PBMC cytokine data from healthy donors, SpA and RA patients derived by Boolean gating, the percentage of each cytokine is represented in relation to the total number of cytokine-producing CD4 T cells. All statistical analysis (except **g**, Friedman's) determined by Kruskal–Wallis test with p-values calculated using Dunn's multiple comparisons test

mononuclear cells by in vitro explant culture. The flow cytometry gating strategy used to identify ILCs in blood and synovial tissue cultures is shown in Supplementary Fig. 5. Lineage-negative, IL-7R⁺ cells were indeed present in the synovial tissue and produced cytokines upon stimulation (Fig. 3a). The predominant ILC population in spondyloarthritis tissue was C-KIT⁺ ILC3s, whilst

in the blood of healthy donors and spondyloarthritis patients ILC1 cells were the predominant subset (Fig. 3b). Figure 3c, d shows that GM-CSF was the most commonly produced cytokine for both ILC1 and ILC3 subsets in the synovial tissue-derived cells. In the synovial cultures, ILC1 showed reduced IFN-γ, while ILC3 showed increased GM-CSF compared to peripheral blood

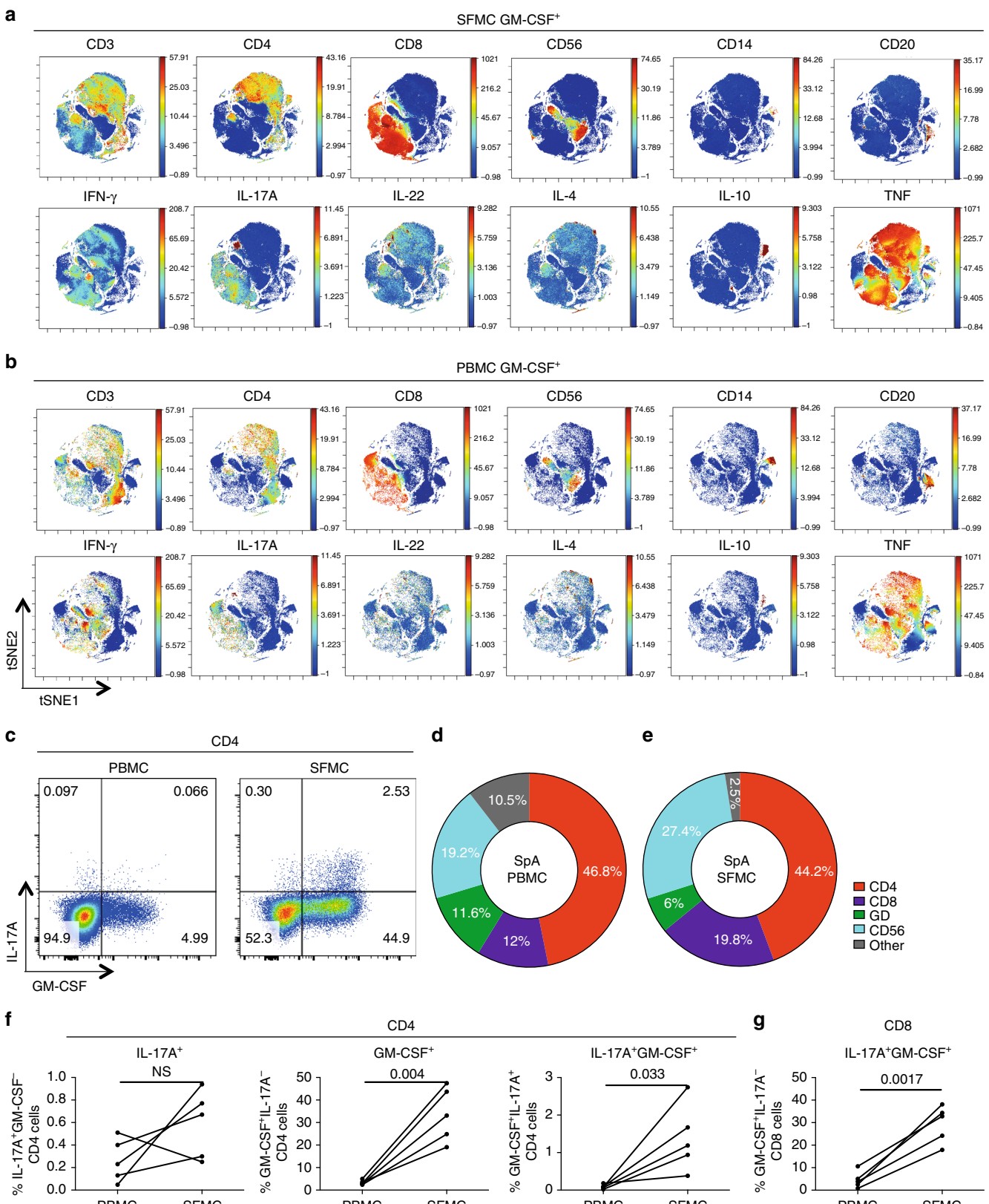

from spondyloarthritis and healthy donors (Supplementary Fig. 5B–E).

**GM-CSF CD4 cells have a distinct transcriptional profile.** Given that our patient and control data suggest that GM-CSF[+]CD4[+] may constitute a distinct functional subset, we next asked if GM-CSF[+] CD4 T cells have a distinct transcription signature compared to classical IFN-γ[+] Th1 cells and IL-17A[+] Th17 cells. In the absence of known surface markers that define these cells we developed a triple-cytokine capture assay (Supplementary Fig. 6A, B shows cytokine capture-stained cells and FACS sorting strategy). This allowed us to purify CD4 T cells producing GM-CSF, IL-17A and IFN-γ both in isolation and in combination. We FACS sorted to high purity (>95%) GM-CSF[+], IL-17A[+] and IFN-γ[+] single-positive cells, IL-17A[+]GM-CSF[+] double-positive cells and cytokine-negative CD45RA[+] naive CD4 T cells, from an initial cohort of 4 healthy donors. Normalised RNAseq gene counts for *IL17A*, *IFNG* and *CSF2* and for the transcription factors *RORC*, *TBX21* and *PTPRC* are shown in Supplementary Fig. 6C. Unbiased hierarchical gene clustering analysis of the 5 sorted subsets showed GM-CSF[+] single positive cells to cluster more closely with IFN-γ[+] single positive cells while IL-17A[+]GM-CSF[+] double-positive cells clustered with IL-17A[+] single positives (Fig. 4a shows mean data, Supplementary Fig. 6E shows all donors). Differential gene expression analysis (compared to CD45RA[+] T cells) showed GM-CSF single positive cells to have a gene expression profile with both unique (Fig. 4a, highlighted in red boxes) and shared features compared to IFN-γ[+] and IL-17A[+] single-positive cells (Fig. 4b). Volcano plots with the top 10 upregulated and downregulated genes in the GM-CSF[+] (Fig. 4c) and IL-17A[+]GM-CSF[+] (Fig. 4d) are shown in addition to network analysis for these two subsets (Fig. 4e). Pathway analysis for the genes common to GM-CSF[+] CD4 cells in Fig. 4f showed altered p53 signalling, glycolysis, Th17 and apoptosis pathways (individual genes used to define each pathway are listed in Supplementary Table 3).

**GPR65 expression is associated with GM-CSF.** Next we asked if any genes selectively expressed in GM-CSF[+] cells (both independently and in combination with IL-17A) might have be potentially pathogenic, integrating our RNA sequencing data with eQTL and GWAS data sets (Fig. 5a). GPR65, a proton-sensing G-protein coupled receptor was the fourth highest ranked gene identified from our bioinformatics ranking analysis and was upregulated in the GM-CSF[+] and IL-17A[+]GM-CSF[+] but not in IL-17A[+] cells. (Fig. 5c). *GPR65* is in complete linkage disequilibrium with a SNP associated with ankylosing spondylitis through GWAS studies[18] (Fig. 5b). Increased expression of *GPR65* in GM-CSF[+] and IL-17A[+]GM-CSF[+] was confirmed by qPCR in a second independent cohort of capture-sorted cells from 3 healthy donors (Supplementary Fig. 6D). Next we studied expression of *GPR65* in pooled Th17 cells from additional spondyloarthritis patients, healthy donors and RA disease controls. We purified ex vivo CD4 IL-17A[+] cells by single cytokine capture from 4 spondyloarthritis, 4 RA and 3 healthy controls. Since we showed that in spondyloarthritis nearly 40% of Th17 cells co-expressed GM-CSF (Fig. 1c), we hypothesised that ex vivo

sorted Th17 cells from spondyloarthritis would have higher expression of *GPR65* compared to the control populations, and this was confirmed in Fig. 5d. We silenced *GPR65* in primary human T cells (confirmed by qPCR, Supplementary Fig. 7A) and observed a significant downregulation of GM-CSF but not IL-17A (Fig. 5e, f). Since GPR65 is a known extracellular pH sensor[30], we cultured isolated primary human CD4 cells in media with a pH of 6.5 and observed a significant increase in GM-CSF production (Fig. 5g).

**IL-7 promotes expansion of GM-CSF in CD4 T cells.** Since IL-7 has been previously been shown to be a driver of GM-CSF in a STAT5 dependent manner in murine models, we sought to investigate the effects of IL-7 on human CD4 cells. Supplementary Fig. 8A, B show that IL-7 enhances the GM-CSF production by CD4 cells, with a doubling of the number of cells expressing this cytokine. IL-7 had no effect on IL-17A, IL-22, IFN-γ, TNF or IL-17A/GM-CSF double-positive cells. Adding recombinant human IL-7 to explant synovial tissue cultures from patients with inflammatory arthritis (Supplementary Fig. 8C) increased both the percentage of GM-CSF[+] CD4 cells and GM-CSF production by ELISA. There was also a statistically significant increase ($p = 0.03$) in the percentage of IL-22[+] cells but no difference in IFN-γ[+], IL-17A[+] and IL-17A[+]GM-CSF+ cells. However, culture in the presence of IL-7 did not increase expression of GPR65 (Supplementary Fig. 7B). Finally, we sought to investigate the downstream target of GM-CSF in ex vivo PBMCs and SFMCs from spondyloarthritis patients by looking at pSTAT5 induction in response to recombinant human GM-CSF. We observed a clear induction of pSTAT5 in CD14 monocytes in the blood and synovial fluid but no effect on CD4[+], CD8[+], CD56[+] and CD20[+] cells (Supplementary Fig. 8D).

**Discussion**

We here show that human spondyloarthritis is characterised by a program of lymphoid GM-CSF production in multiple compartments, including CD4 and CD8 T cells, γδ and innate lymphoid cells (ILC). This overlaps with an expansion of "type 17" cytokine production previously reported for CD4[9] and γδ[20] cells and confirmed by our data. Notably in spondyloarthritis, GM-CSF production is increased both independently of, and in combination with, IL-17A and IFN-γ. This, together with our data showing that CD4 T cells from synovial fluid are further enriched for GM-CSF production, suggests that the GM-CSF programme may be a primary pathogenic process.

We also show increased numbers of lineage-negative IL-7R-positive type 3 innate lymphoid cells (ILC3) in inflamed joints, and further show that ILC3 cells derived from the joint abundantly produce GM-CSF upon in vitro stimulation, albeit after in vitro culture. C-Kit-expressing ILC3 are thought to contribute to the cytokine environment through the production of IL-17A and IL-22 in response to tissue stress[23,31], and have recently been shown to produce GM-CSF in the gut[24]. We show that ILC1s are also present in the joint and show a downregulation of IFN-γ relative to blood. Thus, our data suggest that GM-CSF may be an important mediator in both innate and adaptive immune responses in spondyloarthritis.

---

**Fig. 2** Spondyloarthritis synovial fluid T cells are highly enriched for GM-CSF production ex vivo. CyTOF viSNE clustering analysis of all live GM-CSF[+] events from paired ex vivo spondyloarthritis (SpA) synovial fluid mononuclear cells (SFMC) **a** and PBMC **b**. In the top panels of **a** and **b** CD3, CD4, CD8, CD56, CD14 and CD20 are shown as the third parameter. The bottom panels of **a** and **b** show IFN-γ, IL-17A, IL-22, IL-4, IL-10 and TNF as third parameters. **c** Representative flow cytometry plots, gated on CD4 cells, from matched SpA patient PBMCs and SFMC showing IL-17A and GM-CSF production. **d–e** Flow cytometry data of 5 matched ex vivo SpA PBMC **d** and SFMC **e** samples showing the mean cellular components of the GM-CSF pool. **f** Paired PBMC and SFMC flow cytometry data gated on CD4 showing the percentage of IL-17A[+]GM-CSF[−], IL-17A[−]GM-CSF[+] and IL-17A[+]GM-CSF[+] in SpA patients ($n = 5$, paired *t*-test). **g** Paired PBMC and SFMC flow cytometry data showing the percentage of IL-17A[+]GM-CSF[+] CD8 cells in SpA ($n = 5$, paired *t*-test)

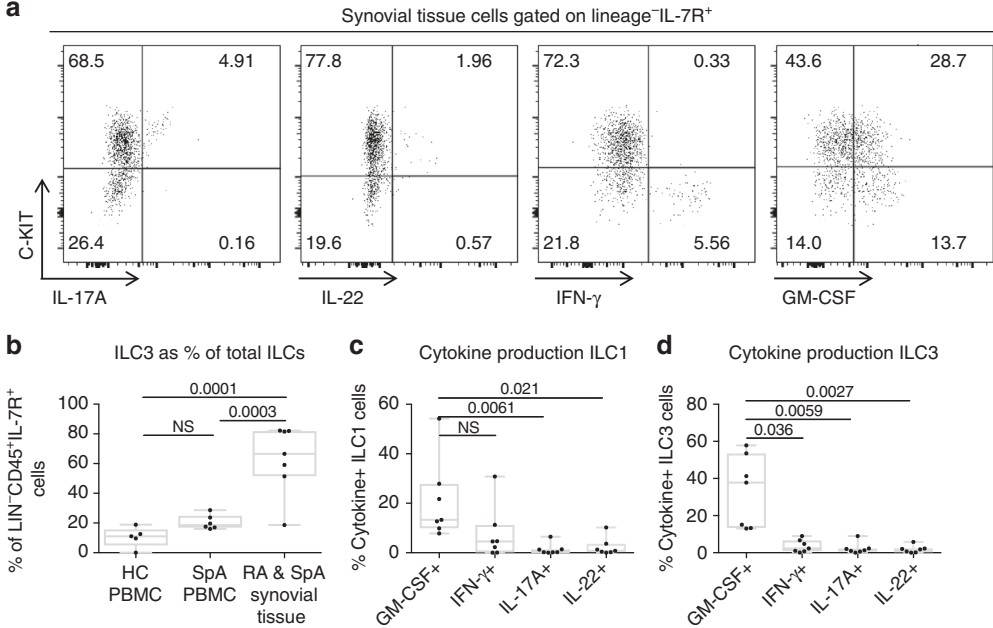

**Fig. 3** Type 3 innate lymphoid cells (ILC3) are enriched in inflammatory arthritis synovial tissue and produce GM-CSF. **a** Representative flow cytometry plots showing cytokine production by synovial ILCs. Lineage negative (Gating strategy shown in Supplementary Fig. 5. Lineage cocktail: CD3, CD5, CD8, CD11b, CD11c, CD19, CD20, CD34, TCR-γδ, CRTH2) CD45+ cells were then gated on IL-7R expression. ILC1 subset defined as Lin−CD45+IL-7R+C-KIT− and ILC3 subset defined as Lin−CD45+IL-7R+C-KIT+. **b** Relative enrichment of ILC3s as percentage of total ILCs in inflammatory arthritis synovial tissue explant cultures (n = 7), compared to ex vivo PBMCs from axial spondyloarthritis (SpA) (n = 6) and healthy control (n = 5) (ANOVA). **c**, **d**. Intracellular expression of GM-CSF IFN-γ, IL-17A and IL-22 in stimulated ILCs plotted against C-KIT expression to identify ILC1 and ILC3 cytokine production (n = 7 patients with inflammatory arthritis, 4 SpA/3 RA) (Friedman's multiple comparison test)

Several lines of evidence have suggested that T cell-derived GM-CSF may be pathogenic in murine and human inflammatory diseases. Neutralization of GM-CSF has been shown to be effective in the Sagakuchi mouse model of spondyloarthritis[15] and in EAE models of neuro-inflamation[16], with a T cell specific-GM-CSF knockout shown to be protected from disease[32]. GM-CSF-mediated immunity has been implicated in several human diseases. Elevated GM-CSF-producing T cells have been shown in the joints of patients with Juvenile Inflammatory Arthritis[33]. Moreover an increased percentage of GM-CSF+ CD4 T cells have been found in the cerebrospinal fluid of patients with multiple sclerosis (MS) compared to non-MS controls[16], and CD4 and CD8 T cells expressing GM-CSF have been shown to be elevated in the peripheral blood of patients with MS[34].

In this study, we observed GM-CSF production both independently and in combination with Th17 subsets. Importantly, we here show that the percentage of IL-17A+GM-CSF+ double-positive cells is greatly increased in the peripheral blood of patients with spondyloarthritis, with further increases in the ex vivo synovial fluid. Th17 cells co-expressing GM-CSF are thought to be particularly pathogenic in animal models of inflammatory disease[17,35,36]. Lee et al.[35] showed pathogenic Th17 cells to be dependent on IL-23 and downstream TGF-β3 signalling, with Csf2 (GM-CSF) one of the key upregulated genes in murine pathogenic Th17 cells. Csf2 was also shown recently to be upregulated in pathogenic Th17 cells isolated from the central nervous system of diseased experimental autoimmune encephalitis (EAE) mice[17].

Our flow cytometry, CyTOF and gene expression data suggest that GM-CSF+ CD4 T cells have a distinct phenotypic and transcriptional profile. Several reports have shown GM-CSF production to be more aligned with a STAT5 rather than a STAT3 programme[16], with IL-7 implicated as a driver of STAT5-mediated GM-CSF-driven inflammation in the mouse EAE

model[37]. Moreover, it has been shown that GM-CSF production by CD4 T cells is influenced by IL-1β[38,39], with GM-CSF in turn acting via monocytes to boost IL-1β production[40]. We recently showed that RORγt inhibition of human Th17 cells in vitro led to suppression of IL-17A+ but not GM-CSF+ CD4 T cells in vitro, suggesting GM-CSF can act independently of RORγt[41]. In the absence of clear surface phenotypic markers to identify GM-CSF+ CD4 T cells and IL-17A+GM-CSF+ double-positive CD4 T cells, our RNA sequencing data represent an important insight into the transcriptional regulation of these cells. We show GM-CSF+ cells to have a transcriptional profile that is distinct from IFN-γ+ and IL-17A+ cells, supporting the concept that GM-CSF+ T cells may be a distinct T-cell phenotype. Gene clustering analysis identified distinct, as well as overlapping groups of differentially expressed genes for both GM-CSF+ cells and IL-17A+GM-CSF+ double-positive cells.

In this study, we observe the percentage of GM-CSF+ CD4 and CD8 T cells in spondyloarthritis to be significantly higher than rheumatoid arthritis (RA), as well as healthy controls. Interestingly, the percentage of GM-CSF+ CD4 cells was especially high in patients receiving anti-TNF therapy, suggesting that this pathway is downstream of TNF. This increase in GM-CSF may be due to the fact that anti-TNF treated patients represent a subset with more aggressive disease requiring the need for biological therapies. Alternatively, the effects of TNF blockade may lead to a loss of negative feedback leading to an increase of GM-CSF. An increase in Th17 numbers after anti-TNF has been previously reported in RA[42] and the two mechanisms may be similar.

Our data would support development of clinical trials of anti-GM-CSF in spondyloarthritis, particularly in view of the safety and efficacy already reported in RA[43,44]. Even though we did not observe an increase in circulating GM-CSF+ lymphocytes in RA, the efficacy of the anti-GM-CSF antibody in this disease may be predominantly due to the local effects of GM-CSF in synovial

tissue, where increased numbers of cells expressing the GM-CSF receptor have been described in both RA and psoriatic arthritis[45]. Additionally, in RA, GM-CSF has been shown to play a function in inflammatory dendritic cell differentiation[46].

We here show that both human GM-CSF-producing CD4 T cells, and GM-CSF+ Th17 cells are characterized by high levels of GPR65 expression. GPR65 is G-protein coupled receptor with an extracellular proton sensing domain[47] associated with ankylosing spondyloarthritis by GWAS[18]. Its function in immune cells

remains unclear. We show culture of primary human CD4 cells in an acidic environment significantly upregulates GM-CSF while silencing of GPR65 reduced GM-CSF. Notably, a low pH environment has been reported in the inflamed synovium[48]. Of interest, Gpr65 expression was upregulated in pathogenic murine Th17 cells, with knockout protecting against EAE[17]. The upregulation of GPR65 in ex vivo sorted spondyloarthritis Th17 cells further validates a function for this receptor in driving pathogenic T cell responses in human disease. Since silencing of GPR65 in

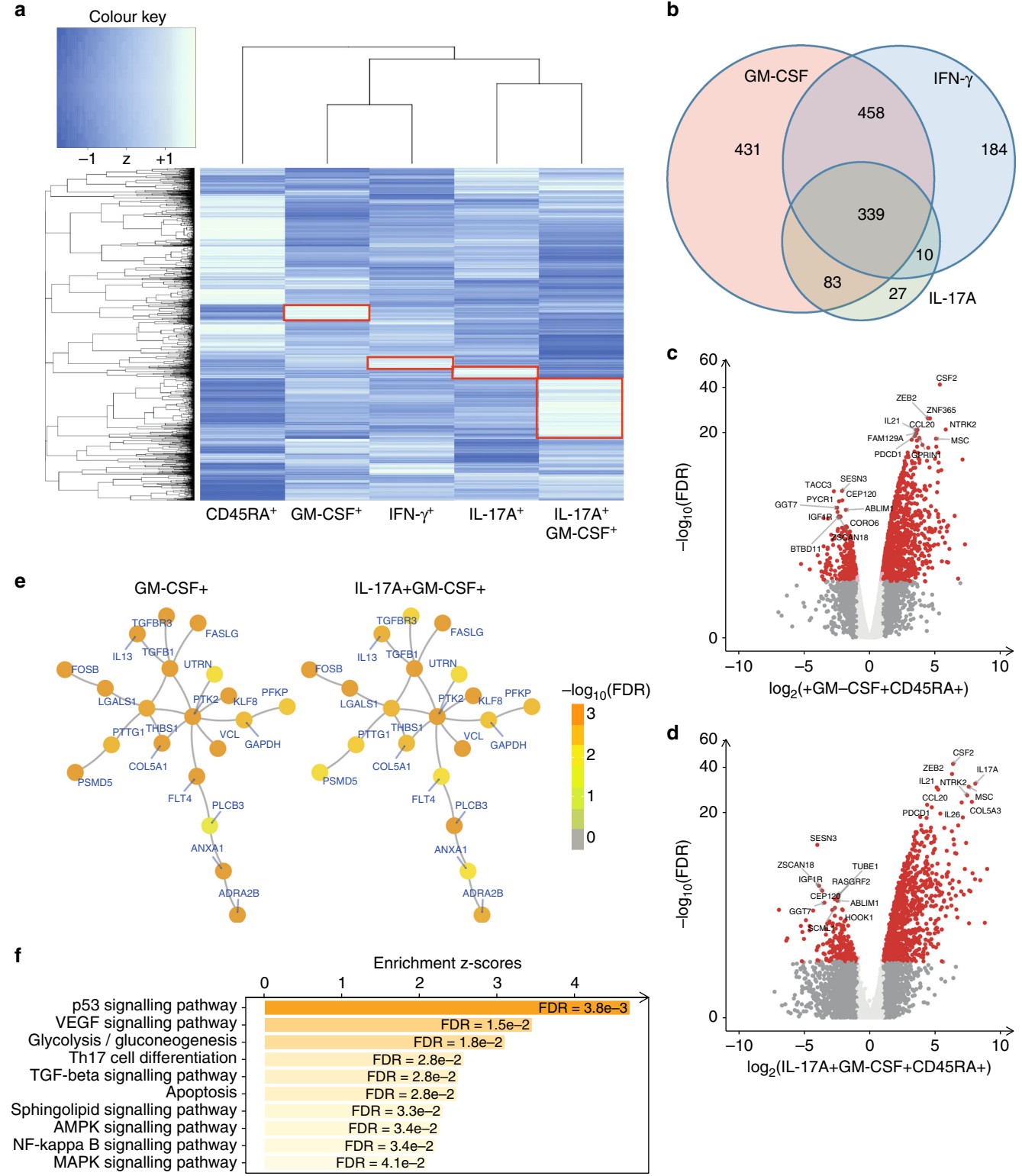

human CD4 cells leads to a significant decrease in GM-CSF, inhibition of GPR65 using small molecules might constitute a promising therapeutic approach, in combination with or as an alternative to current therapies, although the potential human in vivo side effect profile of such agents remains to be determined. Such an approach may potentially preserve the protective function of non GM-CSF secreting Th17 cells[49].

Taken together, our data suggest an important function in human spondyloarthritis for GM-CSF production by multiple lymphocyte populations, including CD4, CD8 and γδ T cells and ILCs, within the context of globally enhanced type 17 immunity. GM-CSF-producing CD4 cells have a distinct transcriptional profile, which includes expression of *GPR65*. Our data provide a strong scientific rationale for clinical trials targeting GM-CSF in spondyloarthritis and additionally identify further targets including GPR65 involved in this likely 'pathogenic' module.

## Methods

**Patient and control recruitment.** Peripheral blood samples were recruited, following informed written consent, from patients attending the Oxford University Hospitals National Health Service Foundation Trust (Ethics reference number 06/Q1606/139, National Health Service, Health Research Authority, South Central—Oxford C Research Ethics Committee). All spondyloarthritis patients met assessment of spondyloarthritis international society (ASAS) criteria for axial spondyloarthritis[28]. Age and sex matched healthy donors and rheumatoid arthritis (RA) patients meeting American college of rheumatology/European league against rheumatism 2010 criteria[50] matched for C-reactive protein were recruited. Patient and control characteristics are shown in Supplementary Table 1. Power calculation was based on 3 pairwise comparisons and observed differences in frequency of Th17 cells from a previous cohort of patients[51]. A minimum sample size of 11 was needed for each group for power of 0.8 and type 1 error rate of 5%.

Synovial fluid from patients with peripheral spondyloarthritis as defined by ASAS criteria[28] or psoriatic arthritis as defined by classification criteria for psoriatic arthritis (CASPAR) criteria[52] were also recruited under the same ethics as the peripheral blood cohort with informed written consent. Additional synovial tissue samples from spondyloarthritis[28] and RA[50] patients were studied (ethics reference number 09/H0606/11, National Health Service, Health Research Authority, South Central—Oxford C Research Ethics Committee). Patient characteristics are shown in Supplementary Table 2.

**Flow cytometry.** The cells were stimulated with phorbol myristate acetate (125 ng/ml; Sigma-Aldrich) and ionomycin (1 µg/ml; Sigma-Aldrich), in the presence of Golgiplug (Brefeldin A) and Golgistop (Monensin) (both from BD Biosciences diluted according to manufacturer's instructions) and incubated for four hours at 37 °C in 5% CO$_2$. Staining antibodies and dilutions shown in Supplementary Table 4. The cells were stained on ice and in the dark for 20 min, then permeabilised using Cytofix/Cytoperm fixation and permeabilization solution (BD Biosciences) for 30 min and stained for intracellular cytokines. Flow cytometry was performed on a BD Fortessa calibrated daily with calibration and tracking beads from BD Biosciences. The data were analysed using FlowJo software (Treestar).

**CyTOF staining.** A volume of 8 µl of 103Rh intercalator (Fluidigm) was added to cells and incubated for a 15 min at 37 °C. Cells were resuspended in 50 µl staining buffer containing surface antibody master mix and incubated for 20 min at room temperature, washed, then re-suspended in 500 µl of fix buffer (Fluidigm-Maxpar) and incubated at room temperature for 20 min. Cells were permeabilise using 2 ml of perm-S-buffer (Fluidigm-Maxpar) and intracellular staining was performed for 30 min at room temperature. All antibodies used are listed in Supplementary Table 5. Finally, cells were suspended in 500 µl of the 191Ir/193Ir intercalator and

incubated overnight. Equalisation beads (Fluidigm-Maxpar) were added at a concentration of 1:10 and the sample was acquired on a CyTOF-1 machine. Analysis of data was carried out using FlowJo software (Treestar) and Cytobank software.

**Cytokine capture.** Fresh lymphocyte cones were obtained from the NHS blood and transplant service and PBMCs obtained by density centrifugation. CD4 T cells were isolated by negative selection (Miltenyi). The next day cells were stimulated with phorbol myristate acetate (125 ng/ml; Sigma-Aldrich) and ionomycin (1 µg/ml; Sigma-Aldrich) for 2 h. Cells were stained on ice for 5 min with IL-17A, IFN-γ and GM-CSF capture antibodies (all Miltenyi biotech) and incubated at 37 °C for 45 min. At 5 min intervals the mixture was resuspended. Cells were then placed on ice to stop cytokine secretion and stained with MACS detection antibodies (IL-17A-PE, IFN- γ -FITC, GM-CSF-Biotin) all at a concentration of 1:25 µl. In addition, cells were also stained with CD45RA-PerCpCy5.5, CD3 BV605, CD4 BV421 and ef780 viability dye followed by with anti-biotin APC secondary. Sorting was carried out using an automated cell sorter BD Aria. Cells were sorted at 4 °C into tubes containing sterile PBS.

**Synovial tissue cultures.** Freshly obtained surgical material was re-suspended in 25 ml of sterile RPRMI immediately. The sample was placed in a large petri dish and cut into 3–4 mm pieces by hand. 20–30 pieces of the sample were placed in a 6-well plate containing 5 ml of warm of RPMI supplemented with 10% human serum containing 100 IU/ml IL-2 (Peprotech) + IL-7 10 ng/ml (Peprotech). Volume of 2.5 ml of media was aspirated every 72 h and replaced with the same volume containing IL-2 100 IU/ml + IL-7 10 ng/ml. After 14 days non-adherent cells were washed off the plate and filtered through a 70 µm filter. Cells were counted, rested overnight and stimulated in the morning for intracellular cytokine staining.

**RNA extraction.** The AllPrep DNA/RNA/miRNA kit (Qiagen, catalogue number 80224) was used for RNA extraction. FACS sorted cells were spun down and re-suspended in 350 µl of RLTplus buffer and transferred to 2 ml tubes. Samples were then stored at −80 °C for batched RNA extraction. Homogenisation of the sample was carried out using the QIAshredder (Qiagen). DNase I was used during the extraction protocol to minimise DNA contamination. RNA was eluted into 35 µl of RNase-free water. The RNA amount was quantified by nano-drop and the RNA samples stored at −80 °C for storage until ready for sequencing.

**RNA sequencing.** RNA sequencing was carried out at the Wellcome Trust Centre for Human Genetics core facility. RNA underwent quality control testing using a bioanalyser followed by cDNA library preparation. Paired end sequencing was performed at 100 base pairs on each side of the DNA fragment on the HiSeq 4000 platform. Total of 20–80 million fragments were sequenced per sample.

**Analytical methods.** Prism version 6 was used for statistical analysis. Bioinformatic analysis of RNA sequencing samples was carried out using validated packages in R. Reads were mapped to human genome reference sequence GRCh37 with tophat2[53], deduplicated using samtools v.1.2[54]. Gene counts were retrieved using htseq-cout[55] and the Ensembl gene annotation. DESeq2[56], an R package, was used for differential gene expression analysis.

**Pathway and network analysis.** Genes common to IL-17A+GM-CSF+ and GM-CSF+ were defined if they, as compared to naive CD45RA+ cells, were differentially expressed (FDR <5%) in both IL-17A+GM-CSF+ cells and GM-CSF+ cells but not in IL-17A+ cells. Pathway analysis was conducted using the XGR software to identify enriched pathways (compiled from KEGG; June 2017). Network analysis was performed using the dnet package to identify a gene network that contains groups of interconnected genes. In brief, the *dNetPipeline* function in the package took as inputs a list of common genes with the significance level (i.e., the per-gene maximum FDR in both cells) and output a gene network (FDR <1%) from a human gene interaction data (compiled from the STRING database; May 2017).

**Fig. 4** GM-CSF positive CD4 cells have a distinct transcriptional profile. RNA was extracted from 5 FACS sorted CD3+CD4+ T-cell populations: CD45RA+ (IFN-γ−IL-17A−GM-CSF−), GM-CSF+, IFN-γ+, IL-17A+ and IL-17A+GM-CSF+ double-positive, of triple cytokine capture-labelled activated PBMCs from four healthy donors, pooled and sequenced on the Illumina HiSeq 4000 platform. **a** Unbiased hierarchical gene clustering analysis of the 5 sorted subsets from the combined data set of 4 donors. **b** Differential gene expression profiles of GM-CSF+, IL-17A+ and IFN-γ+ single-positive cell subsets (compared to CD45RA+ population) showing unique and shared expressed genes. False discovery rate (FDR) for this analysis was set at <0.05%. **c, d** Volcano plots of differential gene expression. Fold changes in the x-axis versus FDR in the y-axis are plotted for genes identified in GM-CSF+ cells **c** and IL-17A+GM- CSF+ cells **d**, Coloured in red are genes with FDR <5% and at least twofold changes (upregulated or downregulated). Highlighted in text are the most significant genes (upregulated or downregulated). **e** Visualisation of the gene network identified by integrative analysis of RNA sequencing data with gene interaction data. This network contains 20 genes that were commonly regulated in both IL-17A+GM-CSF+ cells and GM-CSF+ cells. Illustrations are using the same network layout with nodes colour-coded according to cell-specific FDR in the GM-CSF+ population (left) and the IL-17A+GM-CSF+ population (right). **f** Bar plots of enriched pathways for 391 genes commonly regulated in IL-17A+GM-CSF+ and GM-CSF+ cells. Displayed in the x-axis are enrichment z-scores with FDR labelled inside. Supplementary Table 3 shows the listing of gene members per pathway

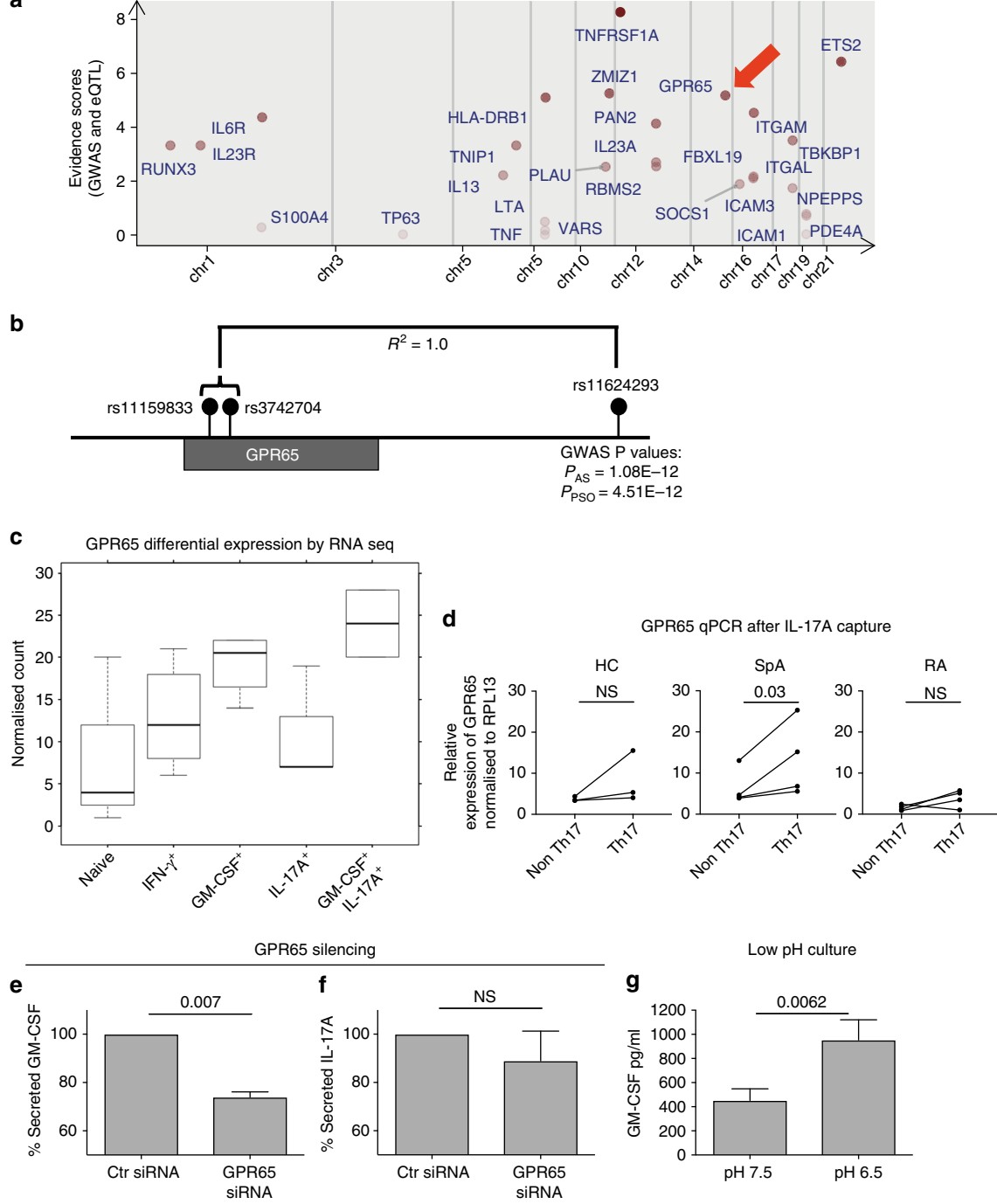

**Fig. 5** GPR65 expression is associated with GM-CSF$^+$ and GM-CSF$^+$IL-17A$^+$ CD4 T cells. **a** Evidence score analysis showing top ranking genes identified in GM-CSF$^+$ and IL-17A$^+$GM-CSF$^+$ cells. Red arrow shows *GPR65* **b** Location of AS and psoriasis-associated *GPR65* single-nucleotide polymorphism rs11624293 in relation the two SNPs identified in eQTL data sets. **c** Expression of *GPR65* in RNA sequencing analysis of the 5 sorted primary T cell populations isolated by multiple cytokine capture ($p = 0.012$, ANOVA). **d** Expression of *GPR65* by qPCR in ex vivo IL-17A$^-$ and IL-17A$^+$ cytokine captured CD4 T cells from patients and controls (paired *t*-tests). GM-CSF **e** and IL-17A **f** measured by ELISA in culture supernatants of primary human CD4 cells in the presence *GPR65* siRNA or control siRNA (mean + SEM, paired *t*-tests). **g** GM-CSF was measured by ELISA in culture supernatants of primary human CD4 cells cultured at a pH of 7.5 or in media acidified to a pH of 6.5 for 72 h (mean + SEM, paired *t*-tests)

**Prioritisation of key genes**. We applied an in-house prioritisation system using random forest plot modelling[57] to generate evidence scores prioritising key genes involved in spondyloarthritis. We utilised GWAS summary data (risk variants and their detected *p*-values) for susceptibility in spondyloarthritis[58,59] and eQTL summary data[60,61] to prioritise the 1643 genes that were differentially expressed in GM-CSF$^+$ CD4 cells from our data set. We prioritised genes based upon the relative weight of evidence of GWAS association and the eQTL significance level linking a SNP to a gene.

**Statistical analysis**. Prism version 6 was used for statistical analysis. When comparing two unpaired groups with normal distribution of data a two-tailed unpaired *t*-test was used for statistical analysis. When comparing two unpaired groups with data that was not normally distributed a Mann–Whitney test was used. For paired data, a paired *t*-test and Wilcoxon matched-pairs signed-rank tests were used for parametric and non-parametric data, respectively. When comparing more than two unpaired groups, a one-way analysis of variance (ANOVA) with Bonferroni's correction was used for parametric data. Where the data were not

normally distributed a Kruskal–Wallis test with Dunn's multiple comparison analysis was used for statistical analysis. For matched data with more than two groups, a one-way ANOVA with Greenhouse-Geisser correction was performed in parametric data sets. In non-parametric data a Friedman test, a Dunn's multiple comparison analysis was performed. Significance was defined as $p \leq 0.05$.

**Data availability**. RNA sequencing data have been deposited in the National Centre for Biotechnology Information gene expression omnibus under the accession number GSE103930. The remainder of the data that support the findings of this study are available on request from the corresponding author. The patient data are not publicly available due to them containing information that could compromise research participant privacy.

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

## Acknowledgements

This research was funded by the Wellcome Trust through a fellowship award to M.H.A.-M., Arthritis Research UK (programme grant 20773, J.C.K., P.B. and I.P.), the National Institute for Health Research (NIHR) Oxford Biomedical Research Centre (P.B. and A.R.). We would like to acknowledge funding by the Structural Genomics Consortium, a registered charity (number 1097737), that receives funds from AbbVie, Bayer Pharma AG, Boehringer Ingelheim, the Canada Foundation for Innovation, Eshelman Institute for Innovation, Genome Canada, Innovative Medicines Initiative (EU/EFPIA) [ULTRA-DD grant no. 115766], Janssen, Merck KGaA Darmstadt Germany, MSD, Novartis Pharma AG, the Ontario Ministry of Economic Development and Innovation, Pfizer, Sao Paulo Research Foundation-FAPESP, Takeda, and the Wellcome Trust [106169/ZZ14/Z]. The views expressed are those of the author(s) and not necessarily those of the National Health Service, the NIHR or the Department of Health.

## Author contributions

M.H.A.-M. Designed and performed experiments, analysed data and wrote the manuscript. L.C. contributed to designing the study and performed experiments. H.F. analysed data. A.R., J.d.W., N.Y. and A.H. performed experiments. I.P. analysed data. B.P.F. contributed to designing the study. D.S., Y.Y. and S.B. performed experiments. K.D., R.G. and B.K. contributed with clinical samples. F.P. and J.C.K. contributed to designing of the study. P.B. contributed to designing of the study and writing the manuscript.

## Additional information

**Competing interests:** Paul Bowness has received research grant funding from the Merck Research Laboratories and Celgene. Mohammad Hussein Al-Mossawi has received unrestricted research grant funding from UCB Pharma. The remaining authors declare no competing financial interests.

