## [Peer Review File · Nature Communications]

Reviewers' comments:

Reviewer #1 (Arthritis, autoimmunity) (Remarks to the Author):

This manuscript identifies an increase of GM-CSF in CD4, CD8 gd, and NK cells in PBMCs and SFMCs from patient with SpA. The gating strategies and the flow and cytof data support these conclusions and show similarities between PB and SF. IL-7 and GPR65 were identified as potential mediators of the increased GM-CSF. There are some major concerns that should be addressed.

- 1) What is the definition of SpA that was used to recruit patients. How many had psoriatic arthritis, ankylosing spondylitis, or enteropathic arthritis. Were there any differences between any of these groups.
- 2) The source of the ST is very unclear. In some places it states the tissue was from SpA patients and in other more vaguely from inflammatory arthritis.
- 3) there are no ST controls for the data in Figure 3. These experiments were performed after a considerable period of culture, which could have biased the results. There are no ST controls. ILC cells were not mentioned elsewhere. IN supplemental figure 5C it states that inflammatory ST was used. What diagnosis?
- 4) RNA-seq was performed on T cell subsets from healthy donors. If I am interpreting panel A correctly this is data from one of these controls. It would be interesting to see how consistent these data are for all four.
- 5) since RNAseq data has been obtained it would be interesting to interrogate further to identify pathways that may be contributing to the increased GM-CSF in addition to GPR65. Data is presented to support the potential role for IL-7. Do the RNAseq support this or another pathway.

Minor concerns:

- 1) was there any statistical significance to the data in Figure 5D?
- 2) the axes in supplemental figure 5A should be added.

Reviewer #2 (RNA-seq, system biology)(Remarks to the Author):

In this manuscript, Al-Mossawi and colleagues have shown significance of GM-CSF producing T cells in development of spondyloarthritis (SpA) by using human patient specimens. The authors showed that GM-CSF+ T cells may constitute a distinct T cell subset. Furthermore, they find a potential role of GPR65 in induction of GM-CSF. Since the potential significance of GM-CSF producing T cells has been already shown in mouse model, as the authors themselves also describe in Introduction, I have to point out that the finding itself is more or less incremental. Nevertheless, I consider that this paper should convey potentially important pieces of information from a clinical point of view. New therapeutic ways may be contemplated based on this paper. Indeed, for this rare disease, the sample size the authors have collected is reasonably high. I would like to suggest some revisions as described below to improve the manuscript.

- 1) The authors described "an expansion of type 3 ILC in inflammatory joint disease" (in Abstract). However, Figure 3 lacks data on the type 3 ILC in healthy donor as a control, leaving some concerns to justify their argument. It is essential to present the results from an appropriate control.
- 2) The authors used eQTL and GWAS data to reveal possible involvement of GPR65 in SpA. However,

details on this part, such as the methods and the evaluations, are poorly documented. The relevance of the results shown in Figure 5A-C is not well discussed. Therefore, I recommend the authors should enrich the descriptions to explain how they have reached the conclusion that GPR65 may be involved, perhaps also utilizing the Supplementary Information section. Indeed, it seems to me that there are other candidates which may be also involved in this disease.

3) The authors claim that GPR65 is silenced in primary human T cells and the silencing reduces the GM-CSF production. In which population of the "primary human T cells", was GPR65 silenced? Did they use GM-CSF+ T cells from patients or healthy donor? Also, evidence of GPR65 silencing (qPCR and/or Western blotting) should be included.

4) It would strengthen the claim of the paper if the authors could explain possible molecular mechanism why silencing of GPR65 led to reduction of GM-CSF. For example, is ligand for GPR65 present in the culture?

5) IL-7 is known to expand the population of GM-CSF+ T cells. How is this result linked to the observed role of GPR65? Would IL-7 induce or activate GPR65? If so, would silencing of GPR65 reduce GM-CSF+ T cell expansion by IL-7? If not, are there two unrelated pathways mediated by GPR65 and IL-7?

6) Downstream consequences of the silencing of GPR65 should be also examined in more comprehensive manner. It seems that extensive RNA Seq experiments using genetic or drug perturbations and/or further in-depth analysis of the obtained data would give a clue?

7) Some discussion regarding the future perspective on the therapeutic strategy of the disease should be added. For example, would not inhibition of the GPR invoke any side effects? How should the new strategy be integrated with the current therapy?

8) Please make sure that all of the data produced for this study have been deposited to the respective public databases under the appropriate access policy.

9) There is no definition of "STMC" (the term is present in Figure 3 and in the last section about induction of GM-CSF by IL-7) in the manuscript. Are they typos of "SFMC"?

Reviewer #3 (Rheumatology, humoral response)(Remarks to the Author):

The manuscript by Al-Mossawi et al describes interesting and novel findings suggesting that a population of GM-CSF-producing T cells is specifically expanded in the peripheral compartment and within the inflamed synovial microenvironment of patients with SpA, compared with HD and RA patients. Additionally, the Authors demonstrate that a large proportion of IL-17A-producing T cells co-express GM-CSF and that these subsets of IL-17 single or IL-17/GM-CSF double-producers are characterised by a unique transcriptional profile using RNASeq. Finally, the Authors identified GPR65 as a candidate regulator of GM-CSF production in T cells as identified by eQTL analysis and siRNA experiments. Although many of the above findings are potentially of significant interest and are supported by in depth and robust FACS, CyTOF and next gen sequencing data, the Authors mostly provide observational findings and fail to provide enough evidence that such observations are or critical pathogenic significance in the pathogenesis of SpA. Additionally, I found surprising that no serious attempt has been made by the Authors to correlate their findings with clinical parameters of disease activity or treatment status making it difficult to appreciate the translational value (if any) of

their work.

I have listed some more specific observations/criticism below.

- 1) In Fig. 1 the Authors suggest that the expansion of IL-17 and IL-17/GM-CSF in CD4+ T cells in the periphery is specific to SpA but not RA. This is in contrast with previous work in early and established RA demonstrating increased circulating IL-17+ T cells (Shen A&R 2009, van Hamburg A&R 2011). I noticed that the RA population selected has rather low/moderate disease activity (mean DAS28 3.3, max 5.3) which might have biased the results.
- 2) The FACS data are essentially confirmed using CyTOF (Fig.2) on matched PBMC and SFMC. I found the CyTOF data slightly underwhelming (somewhat a glorified version of the FACS data in Fig.1) given the much greater potential of CyTOF for a better characterization of specific CD4 subsets. The lack of inclusion of other key T cell surface markers seems a missed opportunity for further clarification of the nature of GM-CSF and IL-17/GM-CSF CD4 T cells. Additionally, the Authors may want to have a double look at the TNFa clustering in Fig.2 A and B which seems remarkably similar.
- 3) Is the observed prevalence of IL-17/GM-CSF+ and GM-CSF+ CD4 T cells in SpA stable over time and/or does it vary with treatment? In particular, the relationship with TNF (is there any evidence of GM-CSF/TNF double-production in CyTOF?) and anti-TNF treatment (i.e. longitudinal sampling and impact on response rate) has not been explored, which is surprising.
- 4) In Fig.3 the Authors present data of GM-CSF production by ILC (mostly ILC1 and ILC3) in synovial tissue of SpA patients. It is puzzling why the Authors do not show data on the GM-CSF and IL-17 production in the tissue T cell compartment but only focus on ILC. A parallel analysis would have helped to investigate whether there is a compartmentalised GM-CSF production in the peripheral vs the joint microenvironment between CD4 T cells and ILC. Also, are they surprised to see no difference in IL-17A production in ILC1 vs ILC3? Finally, the description of the patient population from which the synovial tissue was obtained is completely missing.
- 5) in Fig.4 the Authors present cluster analysis from RNAseq data suggesting that unique transcriptional signatures characterise GM-CSF and IL-17/GM-CSF T cells. The Authors have done a great job in combining triple cytokine capture FACS sorting and RNAseq but Fig.4 as it is presented is again somewhat underwhelming. The list of the top differentially expressed genes between the main subpopulations (either as a list or a series of 2D or 3D volcano plots) would certainly help in understanding the biological significance underlying the observed differences in cytokine production in the different subsets. Also, a pathway analysis might be of help (differences in metabolic pathways would be of significant interest).
- 6) In Figure 5 the Authors described eQTL and GWAS evidence scores and identified GPR65 as a candidate gene upstream of GM-CSF regulation. The bioinformatics processes used are poorly explained. Although the data presented confirmed an enrichment of GPR65 transcript in the GM-CSF and IL-17/GMCSF T cell subsets, the siRNA data performed to claim that GM-CSF production (why ELISA is used here?) is directly regulated through GPR65 is largely inconclusive and lack a proper understanding of the underlying functional mechanisms. Also, are data available that SNPs in GPR65 identified in GWAS and associated with SpA influence the frequency of GM-CSF producing T cells and/or the response to GPR65 targeting by siRNA?
- 7) The final set of data presented in Fig.S5 on the role of IL-7/IL-7R is of interest, but I did not understand the point of adding data on pSTAT5 induction in CD14+ cells stimulated with GM-CSF. A co-culture between autologous CD14+ cells and GM-CSF producing T cells with and without GM-CSF blockade (i.e. using available blocking antibodies) following by a proper analysis of CD14+ phenotypic changes (including M1-M2 polarization) would have made more sense here.

Thank you for asking us to resubmit our manuscript "Human spondyloarthritis is characterized by a programme of lymphocyte GM-CSF production that overlaps type 17 immunity". We are grateful for the very helpful reviewers' comments which we have now addressed, including additional experimental data and analysis as requested. Our detailed response to each comment is shown below:

Reviewer #1 (Arthritis, autoimmunity) (Remarks to the Author):

This manuscript identifies an increase of GM-CSF in CD4, CD8 gd, and NK cells in PBMCs and SFMCs from patient with SpA. The gating strategies and the flow and cytof data support these conclusions and show similarities between PB and SF. IL-7 and GPR65 were identified as potential mediators of the increased GM-CSF. There are some major concerns that should be addressed.

1) What is the definition of SpA that was used to recruit patients. How many had psoriatic arthritis, ankylosing spondylitis, or enteropathic arthritis. Were there any differences between any of these groups.

For the 38 SpA peripheral blood samples studied in Figure 1, all patients fulfilled the ASAS criteria for axial SpA (Rudwaleit et al., Annals of Rheumatic Diseases 2010, 70:25). We have now included the breakdown of the diagnoses in Table S1. 9 had co-existing IBD and 8 psoriasis. We did not observe any significant differences (Kruskall-Wallis) in CD4 PBMC GM-CSF or IL-17A between the groups (fig S2).

For the additional 5 matched peripheral blood and synovial fluid samples studied in Figure 2, three patients had axial SpA, one had B27+ reactive arthritis and one had Psoriatic arthritis (details in new Table S2, left hand panel).

2) The source of the ST is very unclear. In some places it states the tissue was from SpA patients and in other more vaguely from inflammatory arthritis.

Apologies this was not clear. Synovial tissue was obtained from 7 inflammatory arthritis patients undergoing joint replacement surgery, comprising 4 with SpA and 3 with RA. This is now clarified in the text (page 12, line 3), and clinical details shown in new Table S2 (central and right hand panels).

3) there are no ST controls for the data in Figure 3. These experiments were performed after a considerable period of culture, which could have biased the results. There are no ST controls. ILC cells were not mentioned elsewhere. In supplemental figure 5C it states that inflammatory ST was used. What diagnosis?

Good points. Figure 3 was included because firstly it confirms the existence of ILCs in the inflamed joint, but principally to show for the first time that synovial ILCs are major producers of GM-CSF. It is difficult to obtain control tissue from a healthy joint, therefore we are not claiming this to be a disease specific mechanism. However, we have now studied ILC populations in PBMCs from healthy donors and SpA patients and show enrichment of ILC3 in the joint (new Fig. 3b) and increased GM-CSF production from synovial ILC3 (supplementary Fig. S5B-E) compared to these control groups. Whilst we accept the reviewer's comments that the culture may have biased the results - and now include this in the discussion on page 22, paragraph 2, line 4. Of note, new Figure S4 shows that the percentage of GM-CSF production by CD4 cells in the cultured synovial tissue does not differ from that of ex-vivo synovial fluid CD4 cells.

4) RNA-seq was performed on T cell subsets from healthy donors. If I am interpreting panel A correctly this is data from one of these controls. It would be interesting to see how consistent these data are for all four.

Fig. 4A shows the combined dataset from all four donors. This is now clarified in the figure legend.

5) since RNAseq data has been obtained it would be interesting to interrogate further to identify

pathways that may be contributing to the increased GM-CSF in addition to GPR65. Data is presented to support the potential role for IL-7. Do the RNAseq support this or another pathway.

Thank you for this useful suggestion. We have now included both network and pathway analysis of the RNAseq data for the GM-CSF+ CD4 cells as new Figure 4E and F and table S3. This analysis shows enrichment of VEGF, Th17, TGF beta, glycolysis/gluconeogenesis, p53 and apoptosis pathways amongst others. The latter two pathways are indeed downstream of IL-7 signalling.

Minor concerns:

1) was there any statistical significance to the data in Figure 5D?

The P value is 0.012 (ANOVA) and is now included in the legend.

2) the axes in supplemental figure 5A should be added.

These have now been added, thank you. This is now Figure S8.

Reviewer #2 (RNA-seq, system biology)(Remarks to the Author):

In this manuscript, Al-Mossawi and colleagues have shown significance of GM-CSF producing T cells in development of spondyloarthritis (SpA) by using human patient specimens. The authors showed that GM-CSF+ T cells may constitute a distinct T cell subset. Furthermore, they find a potential role of GPR65 in induction of GM-CSF. Since the potential significance of GM-CSF producing T cells has been already shown in mouse model, as the authors themselves also describe in Introduction, I have to point out that the finding itself is more or less incremental. Nevertheless, I consider that this paper should convey potentially important pieces of information from a clinical point of view. New therapeutic ways may be contemplated based on this paper. Indeed, for this rare disease, the sample size the authors have collected is reasonably high. I would like to suggest some revisions as described below to improve the manuscript.

1) The authors described “an expansion of type 3 ILC in inflammatory joint disease” (in Abstract). However, Figure 3 lacks data on the type 3 ILC in healthy donor as a control, leaving some concerns to justify their argument. It is essential to present the results from an appropriate control.

We agree with the reviewer. We have now included in figure 3B (and S5) data showing ILC3 percentages and cytokine production for 5 healthy control PBMC as well as SpA PBMC.

2) The authors used eQTL and GWAS data to reveal possible involvement of GPR65 in SpA. However, details on this part, such as the methods and the evaluations, are poorly documented. The relevance of the results shown in Figure 5A-C is not well discussed. Therefore, I recommend the authors should enrich the descriptions to explain how they have reached the conclusion that GPR65 may be involved, perhaps also utilizing the Supplementary Information section. Indeed, it seems to me that there are other candidates which may be also involved in this disease.

We have now included additional information in the methods section on the bioinformatics analysis used to identify the top-ranking genes of interest (see methods section under new subheading “*Prioritisation of key genes through integrating transcriptome, GWAS and eQTL summary data, and ontology annotation data*”). Out of the top 4 genes identified in our ranking analysis, GPR65 was chosen for further study because it has evidence of immune-mediated function and was independently shown to be a marker of pathogenic Th17 cells by single cell analysis in the Th17 driven murine EAE model (Gaublomme et al., Cell 2016, 163:1400-1412) and is known to be a GWAS hit in Ankylosing Spondylitis.

3) The authors claim that GPR65 is silenced in primary human T cells and the silencing reduces the GM-CSF production. In which population of the “primary human T cells”, was GPR65 silenced? Did they use GM-CSF+ T cells from patients or healthy donor? Also, evidence of GPR65 silencing (qPCR and/or Western blotting) should be included.

The GPR65 was silenced in primary human CD4 cells derived by negative selection from healthy donors. Evidence of GPR65 silencing by qPCR is now shown in Fig S7A.

4) It would strengthen the claim of the paper if the authors could explain possible molecular mechanism why silencing of GPR65 led to reduction of GM-CSF. For example, is ligand for GPR65 present in the culture?

GPR65 is a receptor which senses the extracellular proton ion concentration therefore the ligand is indeed present in the cultures. We have now carried out functional experiments showing a robust induction of GM-CSF secretion from isolated primary human CD4 cells following acidification. This data is shown in figure 5H.

5) IL-7 is known to expand the population of GM-CSF+ T cells. How is this result linked to the observed role of GPR65? Would IL-7 induce or activate GPR65? If so, would silencing of GPR65 reduce GM-CSF+ T cell expansion by IL-7? If not, are there two unrelated pathways mediated by GPR65 and IL-7?

We have carried out additional experiments adding IL-7 to isolated primary human CD4 cells and measuring GPR65 expression by qPCR as requested. We now include these as supplementary figure 7B. The data suggest that IL-7 is acting independently of GPR65 and we have added this to the discussion in page 21, paragraph 1, lines 9-10.

6) Downstream consequences of the silencing of GPR65 should be also examined in more comprehensive manner. It seems that extensive RNA Seq experiments using genetic or drug perturbations and/or further in-depth analysis of the obtained data would give a clue?

We agree that extensive RNA Seq experiments using genetic or drug perturbations would be of interest but feel that this is beyond the scope of the current manuscript. We have however carried out additional experiments showing downstream functional effects of GPR65 acid sensing, and also the effects on GPR65 expression in the presence or absence of IL-7 (figs 5H and S7B).

7) Some discussion regarding the future perspective on the therapeutic strategy of the disease should be added. For example, would not inhibition of the GPR invoke any side effects? How should the new strategy be integrated with the current therapy?

We have now added the following in the discussion on p25, paragraph 1, lines 7-10 "inhibition of GPR65 using small molecules might constitute a promising therapeutic approach, in combination with or as an alternative to current therapies, although the potential human in vivo side effect profile of such agents remains to be determined".

8) Please make sure that all of the data produced for this study have been deposited to the respective public databases under the appropriate access policy.

We will deposit the RNAseq data in the European Genome-phenome Archive.

9) There is no definition of "STMC" (the term is present in Figure 3 and in the last section about induction of GM-CSF by IL-7) in the manuscript. Are they typos of "SFMC"?

Apologies these abbreviations are correct if confusing and have now been clarified. STMC stands for synovial tissue mononuclear cells and SFMC synovial fluid mononuclear cells. In view of the potential confusion, we have removed all mention of STMC from the manuscript and now use "synovial tissue explant cultures"

Reviewer #3 (Rheumatology, humoral response)(Remarks to the Author):

The manuscript by Al-Mossawi et al describes interesting and novel findings suggesting that a population of GM-CSF-producing T cells is specifically expanded in the peripheral compartment and within the inflamed synovial microenvironment of patients with SpA, compared with HD and RA

patients. Additionally, the Authors demonstrate that a large proportion of IL-17A-producing T cells co-express GM-CSF and that these subsets of IL-17 single or IL-17/GM-CSF double-producers are characterised by a unique transcriptional profile using RNASeq. Finally, the Authors identified GPR65 as a candidate regulator of GM-CSF production in T cells as identified by eQTL analysis and siRNA experiments. Although many of the above findings are potentially of significant interest and are supported by in depth and robust FACS, CyTOF and next gen sequencing data, the Authors mostly provide observational findings and fail to provide enough evidence that such observations are of critical pathogenic significance in the pathogenesis of SpA. Additionally, I found surprising that no serious attempt has been made by the Authors to correlate their findings with clinical parameters of disease activity or treatment status making it difficult to appreciate the translational value (if any) of their work.

I have listed some more specific observations/criticism below.

1) In Fig. 1 the Authors suggest that the expansion of IL-17 and IL-17/GM-CSF in CD4+ T cells in the periphery is specific to SpA but not RA. This is in contrast with previous work in early and established RA demonstrating increased circulating IL-17+ T cells (Shen A&R 2009, van Hamburg A&R 2011). I noticed that the RA population selected has rather low/moderate disease activity (mean DAS28 3.3, max 5.3) which might have biased the results.

The referee is entirely correct that some early papers found increased Th17 numbers in RA blood, although this was not replicated by other groups (e.g. Jandus et al., *Arthritis & Rheumatology* 2008, 58(8): 2307). Furthermore, IL-17A inhibition did not show efficacy in RA (Genovese et al., *Annals of Rheumatic Diseases* 2013, 72:863), in marked contrast to SpA. Whilst we agree that our RA disease control group had on average moderate disease activity we show below that DAS for the RA patients does not correlate with IL-17A⁺ and IL17⁺GM-CSF⁺ numbers and thus it is unlikely that our patient inclusion biased results.

2) The FACS data are essentially confirmed using CyTOF (Fig.2) on matched PBMC and SFMC. I found the CyTOF data slightly underwhelming (somewhat a glorified version of the FACS data in Fig.1) given the much greater potential of CyTOF for a better characterization of specific CD4 subsets. The lack of inclusion of other key T cell surface markers seems a missed opportunity for further clarification of the nature of GM-CSF and IL-17/GM-CSF CD4 T cells. Additionally, the Authors may want to have a double look at the TNFα clustering in Fig.2 A and B which seems remarkably similar.

Good points. We have re-analysed the CyTOF data from SpA PBMCs to understand the cell surface expression marker of the single positive IL-17A, GM-CSF and IFN γ subsets and the double positive IL-17A/GM-CSF cells. These data are now represented as a heat map in figure S3E. Our Fig 2A was indeed an error. Many thanks for pointing out this mistake on our part, we have now included the correct data as a new Figure 2A.

3) Is the observed prevalence of IL-17/GM-CSF⁺ and GM-CSF⁺ CD4 T cells in SpA stable over time and/or does it vary with treatment? In particular, the relationship with TNF (is there any evidence of GM-CSF/TNF double-production in CyTOF?) and anti-TNF treatment (i.e. longitudinal sampling and impact on response rate) has not been explored, which is surprising.

Good points. We have now compared the GM-CSF⁺ frequency between patients off and on TNFi therapy, and observe an increase in GM-CSF⁺ CD4 cells in patients on anti-TNFi (fig S2A), now discussed in the main text (p24, paragraph 2, lines 2-5.) We have also re-analysed the data on GM-CSF⁺ and IL-17A⁺GM-CSF⁺ CD4 cells. Even if patients on anti-TNF are excluded we still observe a

statistically significant increase in these two populations in SpA (Kruskal-Wallis) (Fig.S2B middle and right hand panels). CyTOF does indeed show double production of TNF and GM-CSF in CD4, CD8 and CD56 cells in blood and synovial fluid. (Figure 2 A and B). We agree that longitudinal sampling would be valuable and this is on-going.

4) In Fig.3 the Authors present data of GM-CSF production by ILC (mostly ILC1 and ILC3) in synovial tissue of SpA patients. It is puzzling why the Authors do not show data on the GM-CSF and IL-17 production in the tissue T cell compartment but only focus on ILC. A parallel analysis would have helped to investigate whether there is a compartmentalised GM-CSF production in the peripheral vs the joint microenvironment between CD4 T cells and ILC. Also, are they surprised to see no difference in IL-17A production in ILC1 vs ILC3? Finally, the description of the patient population from which the synovial tissue was obtained is completely missing.

Good points. We now include this comparative analysis of CD4 GM-CSF production from blood, synovial fluid and synovial tissue (new Fig. S4).

Lack of IL-17A production from ILC3 populations has been reported by others using single cell RNA seq from the tonsil (Bjorlund et al., Nature Immunology 2016, 17(4):451) and the blood of patients with IBD (Pearson et al., eLIFE 2016, 5:e10066).

Detailed clinical description of patients from which synovial tissue has now been added as Table S2, which also includes data on the patients from whom paired synovial fluid and blood was taken.

5) in Fig.4 the Authors present cluster analysis from RNAseq data suggesting that unique transcriptional signatures characterise GM-CSF and IL-17/GM-CSF T cells. The Authors have done a great job in combining triple cytokine capture FACS sorting and RNAseq but Fig.4 as it is presented is again somewhat underwhelming. The list of the top differentially expressed genes between the main subpopulations (either as a list or a series of 2D or 3D volcano plots) would certainly help in understanding the biological significance underlying the observed differences in cytokine production in the different subsets. Also, a pathway analysis might be of help (differences in metabolic pathways would be of significant interest).

We have now include in figure 4C-D volcano plots with annotation of the top 10 upregulated and downregulated genes in the GM-CSF/IL-17A double positive population and the GM-CSF single positive cells. In panel E of figure 4 we also show a network analysis for the two populations. Pathway analysis for genes common to IL-17A/GM-CSF double positive and GM-CSF single positive cells is now shown in Figure 4F + supplementary table S3).

6) In Figure 5 the Authors described eQTL and GWAS evidence scores and identified GPR65 as a candidate gene upstream of GM-CSF regulation. The bioinformatics processes used are poorly explained. Although the data presented confirmed an enrichment of GPR65 transcript in the GM-CSF and IL-17/GMCSF T cell subsets, the siRNA data performed to claim that GM-CSF production (why ELISA is used here?) is directly regulated through GPR65 is largely inconclusive and lack a proper understanding of the underlying functional mechanisms. Also, are data available that SNPs in GPR65 identified in GWAS and associated with SpA influence the frequency of GM-CSF producing T cells and/or the response to GPR65 targeting by siRNA?

A more detailed explanation of the bioinformatics has now been included in the methods section under the specific sub-heading "*Prioritisation of key genes through integrating transcriptome, GWAS and eQTL summary data, and ontology annotation data*" on page 29. Of the top 4 genes identified in our ranking analysis, GPR65 was chosen for further study because it was independently shown to be a marker of pathogenic Th17 cells by single cell analysis in the Th17-driven murine EAE model (Gaublomme et al., Cell 2016, 163:1400-1412). ELISA is used to study the functional effect of GPR65 silencing as that would be more relevant in the disease setting. We have now also carried out additional experiments to show that culturing cells in the presence of an increase of the natural ligand of GPR65 (H⁺ ions) leads to an increase in GM-CSF production from isolated human CD4 cells (Figure 5, panel H). There is no data available on GPR65 SNPs influencing GM-CSF+ cells and this would require a large-scale study of typed patients to have adequate power.

7) The final set of data presented in Fig.S5 on the role of IL-7/IL-7R is of interest, but I did not

understand the point of adding data on pSTAT5 induction in CD14+ cells stimulated with GM-CSF. A co-culture between autologous CD14+ cells and GM-CSF producing T cells with and without GM-CSF blockade (i.e. using available blocking antibodies) following by a proper analysis of CD14+ phenotypic changes (including M1-M2 polarization) would have made more sense here.

We have now performed and show below the experiments requested. [redated]

Reviewers' comments:

Reviewer #1 (Remarks to the Author):

The authors have been responsive to the review. There are still concerns.

1) In the methods the authors need to include the fact that all the SpA patients fulfilled ASAS criteria. Also the criteria for those who donated SF and ST should be documented. The criteria used to diagnose RA and PsA should be mentioned.

2) Figure 4A appears to present the means of 4 samples in each group. the results for each individual sample should be presented to allow the reader to understand the variability. Also the similarity of clusters mentioned in the results is not apparent to me. this should be clarified.

3) I understand the data in Figure 4E but relevance of why the dots are positioned the way they are is not apparent. Should they be connected?

Reviewer #2 (Remarks to the Author):

I believe this manuscript has been significantly improved by the series of extensive analyses. At least, I found all of the concerns which I raised in the previous round of the review have been satisfactorily addressed. Indeed, it is intriguing to further analyze the biological consequences of the increased activity of GPR65. I sincerely hope the authors continue their efforts to further elucidate the mechanisms to obtain a more comprehensive view on the molecular etiology of this disease, spondyloarthritis.

Reviewer #3 (Remarks to the Author):

The manuscript has certainly improved and remains of significant interest but in my opinion it still lacks clear mechanistic or clinical evidence that

i) the identified subset of GM-CSF and IL-17/GM-CSF CD4 T cells play a pathogenic role in SpA. The new data in Fig S2 showing an increase in the prevalence of GM-CSF CD4 T cells in patients treated with anti-TNF could rather suggest that they are protective (unless these are non-responders, of course);

ii) it represents a truly unique subset of CD4 T cells and not a particular activated state acquired in the local chronic inflammatory microenvironment (this is more important semantically than biologically though, but the Authors make a major point of it);

iii) GPR65 is uniquely and directly functionally related to GM-CSF. The new experiments with H⁺ ions are interesting but inconclusive as acidification per se can alter glycolysis and lactate levels which has specific transporters on CD4 T cells able to modulate, among others, IL-17 responses (no data on GM-CSF exist as far as I know but the glycolytic pathway interestingly emerged in the pathways analysis suggested to the Authors).

Thank you for asking us to resubmit; we thank the reviewers for their further comments. We address these in turn below.

Reviewer #1 (Remarks to the Author):

The authors have been responsive to the review. There are still concerns.

1) In the methods the authors need to include the fact that all the SpA patients fulfilled ASAS criteria. Also the criteria for those who donated SF and ST should be documented. The criteria used to diagnose RA and PsA should be mentioned.

Many thanks for raising this point. In "Patient and Control Recruitment methods page 5, line 2" we now include the statement that "All SpA patients met ASAS criteria for axial SpA²⁸. The criteria used to define SpA, RA and PsA are now included "RA patients meeting ACR/EULAR 2010 criteria⁵⁰" "psoriatic arthritis as defined by CASPAR criteria⁵²". Patient and control characteristics are now shown in Supplementary Tables 1 and 2.

2) Figure 4A appears to present the means of 4 samples in each group. the results for each individual sample should be presented to allow the reader to understand the variability. Also the similarity of clusters mentioned in the results is not apparent to me. this should be clarified.

A heatmap showing the results for each individual sample are now included in a new figure (supplementary figure 6E). The unique clusters for each of the cytokine-positive subsets are now highlighted in red boxes in figure 4A.

3) I understand the data in Figure 4E but relevance of why the dots are positioned the way they are is not apparent. Should they be connected?

Thank you for pointing this out. Unfortunately, the PDF conversion software caused the lines connecting the genes in figure 4A to disappear, this has now been resolved on the new figure.

Reviewer #2 (Remarks to the Author):

I believe this manuscript has been significantly improved by the series of extensive analyses. At least, I found all of the concerns which I raised in the previous round of the review have been satisfactorily addressed. Indeed, it is intriguing to further analyze the biological consequences of the increased activity of GPR65. I sincerely hope the authors continue their efforts to further elucidate the mechanisms to obtain a more comprehensive view on the molecular etiology of this disease, spondyloarthritis.

We thank the reviewer for their assistance and their positive comments.

Reviewer #3 (Remarks to the Author):

The manuscript has certainly improved and remains of significant interest but in my opinion it still lacks clear mechanistic or clinical evidence that

i) the identified subset of GM-CSF and IL-17/GM-CSF CD4 T cells play a pathogenic role in SpA. The new data in Fig S2 showing an increase in the prevalence of GM-CSF CD4 T cells in patients treated with anti-TNF could rather suggest that they are protective (unless these are non-responders, of course);

We thank the reviewer for their comments. Whilst we accept that it is impossible to prove that these GM-CSF and IL-17/GM-CSF CD4 T cells play a pathogenic role in human SpA, we discuss evidence supporting pathogenicity; 1) increased presence of these cells in inflamed joints (discussion lines 5-6 "This, together with our data showing that CD4 T cells from synovial fluid are further enriched for GM-CSF production, suggest that it may be a primary pathogenic process."

2) We also discuss in discussion paragraph 3 direct evidence of pathogenicity in murine models.

We don't believe it is possible to draw firm conclusions about the increased prevalence of GM-CSF-positive CD4s in the anti-TNF treated cohort. This increase in GM-CSF could also be because the anti-TNF treated patients represent a subset with more aggressive disease requiring the need for biological therapies. Alternatively, the effects of TNF blockade may lead to a loss of negative feedback regulatory loops which may lead to an increase in GM-CSF. An increase in Th17 numbers after anti-TNF has been previously reported in RA and the two mechanisms may be similar. These points have now been included in our discussion on page 13 paragraph 2.

ii) it represents a truly unique subset of CD4 T cells and not a particular activated state acquired in the local chronic inflammatory microenvironment (this is more important semantically than biologically though, but the Authors make a major point of it);

We thank the reviewer for these comments. We have now altered the manuscript including the subtitle which now reads that "*GM-CSF positive CD4 T have a distinct transcriptional profile*", therefore reflecting the data we have rather than making claims about a new subset of T cells. We did however observe the distinct transcription profile in CD4 GM-CSF-single-positive cells in the *peripheral blood* of healthy donors and therefore think it unlikely that we are observing a particular activation state acquired in a local chronic inflammatory microenvironment.

iii) GPR65 is uniquely and directly functionally related to GM-CSF. The new experiments with H⁺ ions are interesting but inconclusive as acidification per se can alter glycolysis and lactate levels which has specific transporters on CD4 T cells able to modulate, among others, IL-17 responses (no data on GM-CSF exist as far as I know but the glycolytic pathway interestingly emerged in the pathways analysis suggested to the Authors).

We thank the reviewer for these comments. We agree that the acidification experiments on their own are inconclusive, but taken together with the GPR65 silencing data are supportive of the proposal that GPR65 may modulate GM-CSF by sensing extracellular pH. In our discussion, we state that "We show culture of primary human CD4 cells in an acidic environment significantly upregulates GM-CSF while silencing of *GPR65* reduced GM-CSF". We do not make claims that the acid culture system is specifically working via GPR65.